# Reducing Class Collapse in Metric Learning with Easy Positive Sampling

## Abstract

Metric learning seeks perceptual embeddings where visually similar instances are close and dissimilar instances are apart, but learned representation can be sub-optimal when the distribution of intra-class samples is diverse and distinct sub-clusters are present. We theoretically prove and empirically show that under reasonable noise assumptions, prevalent embedding losses in metric learning, e.g., triplet loss, tend to project all samples of a class with various modes onto a single point in the embedding space, resulting in class collapse that usually renders the space ill-sorted for classification or retrieval. To address this problem, we propose a simple modification to the embedding losses such that each sample selects its nearest same-class counterpart in a batch as the positive element in the tuple. This allows for the presence of multiple sub-clusters within each class. The adaptation can be integrated into a wide range of metric learning losses. Our method demonstrates clear benefits on various fine-grained image retrieval datasets over a variety of existing losses; qualitative retrieval results show that samples with similar visual patterns are indeed closer in the embedding space.

## 1 Introduction

Metric learning aims to learn an embedding function to lower dimensional space, in which semantic similarity translates to neighborhood relations in the embedding space (Lowe, 1995). Deep metric learning approaches achieve promising results in a large variety of tasks such as face identification (Chopra et al., 2005; Taigman et al., 2014; Sun et al., 2014), zero-shot learning (Frome et al., 2013), image retrieval (Hoffer & Ailon, 2015; Gordo et al., 2016) and fine-grained recognition (Wang et al., 2014).

In this work we investigate the family of losses which optimize for an embedding representation that enforces that all modes of intra-class appearance variation project to a single point in embedding space. Learning such an embedding is very challenging when classes have a diverse appearance. This happens especially in real-world scenarios where the class consists of multiple modes with diverse visual appearance. Pushing all these modes to a single point in the embedding space requires the network to memorize the relations between the different class modes, which could reduce the generalization capabilities of the network and result in sub-par performance.

Recently researchers observed that this phenomena, where all modes of class appearance "collapse" to the same center, occurs in case of the classification SoftMax loss (Qian et al.). They proposed a multi-center approach, where multiple centers for each class are used with the SoftMax loss to capture the hidden distribution of the data to solve this issue. Instead of using SoftMax, it was shown that triplet loss may offer some relief from class collapsing (Wang et al., 2014) and this is certainly true in noise-free environments. However, in this paper, we show that in real-world conditions with modest noise assumptions, triplet and other metric learning loss yet suffer from class collapse.

Rather than refine the loss, we argue the key lies in an improved strategy for sampling and selecting the examples. Early work (Malisiewicz & Efros, 2008) proposed per-exemplar distance representation as a means to overcome class collapsing; inspired by this we introduce a simple sampling method to select positive pairs of training examples. Our method can be combined naturally with other popular sampling methods. In each training iteration, given an anchor and a batch of samples in the same category, our method selects the closest sample to the anchor in the current embedding space as the

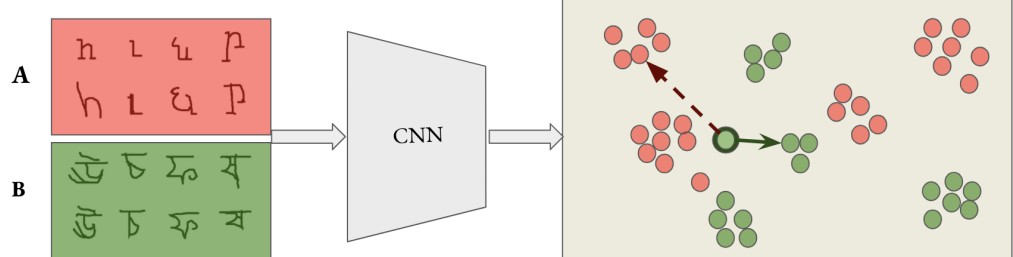

Figure 1: Given an anchor (circle with dark ring), our approach samples the closest positive example in the embedding space as the positive element. This results in pushing the anchor only towards the closest element direction (green arrow), which allows the embedding to have multiple clusters for each class.

positive sample. The metric learning loss is then computed based on the anchor and its positive paired sample.

We demonstrate the class-collapsing phenomena on a real-world dataset, and show that our method is able to create more diverse embedding which result in a better generalization performance. We evaluate our method on three standard zero-shot benchmarks: CARS196 (Krause et al., 2013), CUB200-2011 (Wah et al., 2011) and Omniglot (Lake et al., 2015). Our method achieves a consistent performance enhancement with respect to various baseline combinations of sampling methods and embedding losses.

## 2 RELATED WORK

**Sampling methods.** Designing a good sampling strategy is a key element in deep metric learning. Researchers have been proposed sampling methods when sampling both the negative examples as well as the positive pairs. For negative samples, studies have focused on sampling hard negatives to make training more efficient (Simo-Serra et al., 2015; Schroff et al., 2015; Wang & Gupta, 2015; Oh Song et al., 2016; Parkhi et al., 2015). Recently, it has been shown that increasing the negative examples in training can significantly help unsupervised representation learning with contrastive losses (He et al., 2020; Wu et al., 2018; Chen et al., 2020).

Besides negative examples, methods for sampling hard positive examples have been developed in classification and detection tasks (Loshchilov & Hutter, 2015; Shrivastava et al., 2016; Arandjelovic et al., 2016; Cubuk et al., 2019; Singh & Lee, 2017; Wang et al., 2017). The central idea is to perform better augmentation to improve the generalization in testing (Cubuk et al., 2019). Apart from learning with SoftMax classification, Arandjelovic et al. (2016) propose to perform metric learning by assigning the near instance from the same class as the positive instance. As the positive training set is noisy in their setting, this method leads to features invariant to different perspectives. Different from this approach, we use this method in a clean setting, where the purpose is to get the opposite result of maintaining the inner-class modalities in the embedding space. Xuan et al. (2020) also propose to use this positive sampling method with respect to the N-pair loss (Sohn, 2016) in order to relax the constraints of the loss on the intra-class relations. From a theoretic perspective, we prove that in a clean setting this relaxation is redundant for other popular metric losses like the triplet loss (Chechik et al., 2010) and margin loss (Wu et al., 2017). We formulate the noisy-environment setting and prove that in this case the triplet and margin losses also suffer from class-collapsing and using our purpose positive sampling method optimizes for solutions without class-collapsing. We also provide an empirical study that supports the theoretic analysis.

**Noisy label problem.** Learning with noisy labels is a practical problem when applied to the real world (Scott et al., 2013; Natarajan et al., 2013; Shen & Sanghavi, 2019; Reed et al., 2014; Jiang et al., 2017; Khetan et al., 2017; Malach & Shalev-Shwartz, 2017), especially when training with large-scale data (Sun et al., 2017). One line of work applies a data-driven curriculum learning approach where the data that are most likely labeled correctly are used for learning in the beginning, and then harder data is taken into learning during a later phase (Jiang et al., 2017). Researchers have also tried on to apply the loss only on easiest top k-elements in the batch, determine by lowest current loss (Shen

& Sanghavi, 2019). Inspired by these works, our method focuses on selecting only the top easiest positive relations in the batch.

**Beyond memorization.** Deep networks are shown to be extremely easy to memorize and over-fit to the training data (Zhang et al., 2016; Recht et al., 2018; 2019). For example, it is shown the network can be trained with randomly assigned labels on the ImageNet data, and obtain $100\%$ training accuracy if augmentations are not adopted. Moreover, even the CIFAR-10 classifier performs well in the validation set, it is shown that it does not really generalize to new collected data which is visually similar to the training and validation set (Recht et al., 2018). In this paper, we show that when allowing the network the freedom not to have to learn inner-class relation between different class modes, we can achieve much better generalization, and the representation can be applied in a zero-shot setting.

## 3 PRELIMINARIES

Let $X = \{x_1, .., x_n\}$ be a set of samples with labels $y_i \in \{1, .., m\}$. The objective of metric learning is to learn an embedding $f(\cdot, \theta) \to \mathbb{R}^k$, in which the neighbourhood of each sample in the embedding space contains samples only from the same class. One of the common approaches for metric learning is using embedding losses in which at each iteration, samples from the same class and samples from different classes are chosen according to same sampling heuristic. The objective of the loss is to push away projections of samples from different classes, and pull closer projections of samples from a same class. In this section, we introduce a few popular embedding losses.

**Notation:** Let $x_i, x_j \in X$, define: $D_{x_i,x_j}^f = \| f(x_i) - f(x_j) \|^2$. In cases where there is no ambiguity we omit $f$ and simply write $D_{x_i,x_j}$. We also define the function $\delta_{x_i,x_j} = \begin{cases} 1 & \text{if } y_i = y_j \\ 0 & \text{otherwise} \end{cases}$. Lastly, for every $a \in \mathbb{R}$, denote $(a)_+ := max(a, 0)$.

The Contrastive loss (Hadsell et al.) takes sample embeddings and pushes the samples from the different classes apart and pulls samples from the same class together.

$$\mathcal{L}_{con}^f(x_i, x_j) = \delta_{x_i,x_j} \cdot D_{x_i,x_j}^f + (1 - \delta_{x_i,x_j}) \cdot (\alpha - D_{x_i,x_j}^f)_+$$

Here $\alpha$ is the margin parameter which defines the desired minimal distance between samples from different classes.

While the Contrastive loss imposes a constraint on a pair of samples, the Triplet loss (Chechik et al., 2010) functions on a triplet of samples. Given a triplet $x_a, x_p, x_n \in X$, the triplet loss is defined by

$$\mathcal{L}_{trip}^f(x_a, x_p, x_n) = \delta_{x_a,x_p} \cdot (1 - \delta_{x_a,x_n}) \cdot (D_{x_a,x_p}^f - D_{x_p,x_n}^f + \alpha)_+$$

The Margin loss (Wu et al., 2017) aims to exploit the flexibility of Triplet loss while maintaining the computational efficiency of the Contrastive loss. This is done by adding a variable which determines the boundary between positive and negative pairs; given an anchor $x_a \in X$ the loss is defined by

$$\mathcal{L}_{margin}^{f,\beta}(x_a, x) = \delta_{x_a,x} \cdot (D_{x_a,x}^f - \beta_{x_a} + \alpha)_+ + (1 - \delta_{x_a,x}) \cdot (\beta_{x_a} - D_{x_a,x}^f + \alpha)_+$$

## 4 CLASS-COLLAPSING

The contrastive loss objective is to pull all the samples with the same class to a single point in the embedding space. We call this the *Class-collapsing* property. Formally, an embedding $f : X \to \mathbb{R}^m$ has the class-collapsing property, if there exists a label $y$ and a point $p \in \mathbb{R}^m$ such that $\{f(x_i) | \quad y_i = y\} = \{p\}$.

### 4.1 EMBEDDING LOSSES OPTIMAL SOLUTION

It is easy to see that an embedding function $f$ that minimizes:

$$\mathbb{O}_{con}(f) = \frac{1}{n^2} \left( \sum_{x_i,x_j \in X} \mathcal{L}_{con}^f(x_i, x_j) \right)$$

has the class-collapsing property with respect to all classes. However, this is not necessarily true for the Triplet loss and the Margin loss.

For simplification for the rest of this subsection we will assume there are only two classes. Let $A \subset X$ be a subset of elements such that all the elements in $A$ belongs to one class and all the element in $A^c$ belong to the other class.

Recall some basic set definitions.

**Definition 1.** For all sets $Y, Z \subset \mathbb{R}^m$ define:

1. The diameter of $Y$ is defined by:
$$diam(Y) = \sup\{\|y - z\| \ |y, z \in Y\}$$

2. The distance between Y and Z is:
$$\|Y - Z\| = \inf\{\|y - z\| \ |y \in Y, z \in Z\}$$

It is easy to see that if $f : X \to \mathbb{R}^m$ is an embedding, such that $diam(f(A)) < 2 \cdot \alpha + \|f(A) - f(B)\|$, then:

$$\mathbb{O}_{trip}(f) = \frac{1}{n^3} \left( \sum_{x_i, x_j, x_k \in X} \mathcal{L}_{trip}^f(x_i, x_j, x_k) \right) = 0.$$

Moreover, fixing $\beta_{x_i} = \alpha$ for every $x_i \in X$, then:

$$\mathbb{O}_{margin}(f, \beta) = \frac{1}{n^2} \left( \sum_{x_i, x_j \in X} \mathcal{L}_{margin}^{f,\beta}(x_i, x_j) \right) = 0.$$

It can be seen that indeed, the family of embedding which induce the global-minimum with respect to the Triplet loss and the Margin loss, is rich and diverse. However, as we will prove in the next subsection, this does not remain true in a noisy environment scenario.

## 4.2 NOISY ENVIRONMENT ANALYSIS

For simplicity we will also discuss in this section the binary case of two labels, however this could be extended easily to the multi-label case.

The noisy environment scenario can be formulated by adding uncertainty to the label class. More formally, let $Y = \{Y_1, .., Y_n\}$ be a set of independent binary random variables. Let $A_1, .., A_t \subset X$, $0.5 < p < 1$ such that: $|A_j| = \frac{n}{t}$ and

$$\mathbb{P}(Y_i = k) = \begin{cases} p & x_i \in A_k \\ q' := \frac{1-p}{t-1} & x_i \notin A_k \end{cases}$$

We can also reformulate $\delta$ as a binary random variable such that:

$$\overline{\delta}_{Y_i, Y_j} := \mathbf{1}_{Y_i = Y_j}$$

For the Triplet loss define:

$$\mathbb{E}\mathcal{L}_{trip}^f(x_i, x_j, x_k) = \mathbb{E}\left(\overline{\delta}_{Y_i, Y_j} \cdot (1 - \overline{\delta}_{Y_i, Y_k})\right) \cdot \left(D_{x_i, x_j}^f - D_{x_i, x_k}^f + \alpha\right)_+.$$

We are searching for an embedding function which minimize

$$\mathbb{E}\mathbb{O}_{trip}(f) = \frac{1}{n^3} \sum_{x_i, x_j, x_k \in X} \mathbb{E}\mathcal{L}_{trip}^f(x_i, x_j, x_k)$$

**Theorem 1.** *Let $f : O \to \mathbb{R}^m$ be an embedding, which minimize $\mathbb{E}\mathbb{O}_{trip}(f)$, then $f$ has the class-collapsing property with respect to all classes.*

Similarly, we can define:

$$\mathbb{E}\mathcal{L}^f_{margin}(x_i, x_j) = \mathbb{E}\bar{\delta}_{Y_i, Y_j} \cdot (D^f_{x_i, x_j} - \beta_{x_i} + \alpha)_+ + \mathbb{E}(1 - \bar{\delta}_{Y_i, Y_j}) \cdot (\beta_{x_i} - D^f_{x_i, x_j} + \alpha)_+$$

**Theorem 2.** *Let* $f : O \rightarrow \mathbb{R}^m$ *be an embedding, which minimize*

$$\mathbb{E}\mathbb{O}_{margin}(f, \beta) = \frac{1}{n^2} \sum_{x_i, x_j \in X} \mathbb{E}\mathcal{L}^f_{margin}(x_i, x_j),$$

*then* $f$ *has the class-collapsing property with respect to all classes.*

The proof of the last two theorems can be find in Appendix A.

In conclusion, although theoretically in clean environments the Triplet loss and Margin loss should allow more flexible embedding solutions, this does not remain true when noise is considered. On a real-world data, where mislabeling and ambiguity can be usually be found, the optimal solution with respect to both these losses becomes degenerate.

### 4.3 EASY POSITIVE SAMPLING (EPS)

Using standard embedding losses for metric learning can result in an embedding space in which visually diverse samples from the same class are all concentrated in a single location in the embedding space. Since the standard evaluation and prediction method for image retrieval tasks are typically based on properties of the K-nearest neighbours in the embedding space, the class-collapsing property is a side-effect which is not necessarily in order to get optimal results. In the next section, we will show experimental results, which support the assumption that complete class-collapsing can hurt the generalization capability of the network.

To address the class-collapsing issue we propose a simple method for sampling, which results in weakening the objective penalty on the inner-class relations, by applying the loss only on the closest positive sample. Formally we define the EPS sampling in the following way; given a mini-batch with $N$ samples, for each sample $a$, let $C_a$ be the set of elements from the same class as $a$ in the mini-batch, we choose the positive sample $p_a$ to be

$$\arg\min_{t \in C_a}(\|f(t) - f(a)\|)$$

For negative samples $n_a$ we can choose according to various options. In this paper we use the following methods: **(a)** Choosing randomly from all the elements which are not in $C_a$. **(b)** Using distance sampling (Wu et al., 2017). **(c)** semi-hard sampling (Schroff et al., 2015),**(d)** MS hard-mining sampling (Wang et al., 2019). We then apply the loss on the triplets $(a, p_a, n_a)$. Using such sampling changes the loss objective such that instead of pulling all samples in the mini-batch from the same class to be close to the anchor, it only pulls the closest sample to the anchor (with respect to the embedding space) in the mini-batch, see Figure 1.

In Appendix B, we formalize this method in the noisy environment framework. We prove (Claim 1,2) that every embedding which has the class collapsing property is *not* a minimal solution with respect to both the margin and the triplet loss with the easy positive sampling. Furthermore, in Claim 3,4 we prove that the objective of the losses with EPS on tuples/triplets is to push away every element (including positive elements), that is not in the k-closest elements to the anchor, where k is determined by the noise level $p$. Therefore, if we apply the EPS method on a mini-batch which has small numbers of positive elements from each modality, in such case adding the EPS to the losses not only relax the constraints on the embedding, allowing the embedding to have multiple inner-clusters. It also optimizes the embedding to have this form.

## 5 EXPERIMENTS

We test our EPS method on image retrieval and clustering datasets. We evaluate the image retrieval quality based on the recall@k metric (Jégou et al., 2011) , and the clustering quality by using the normalized mutual information score (NMI) (Manning et al., 2008). The NMI measures the quality of clustering alignments between the clusters induced by the ground-truth labels and clusters induced by applying clustering algorithm on the embedding space. The common practice to choose the NMI

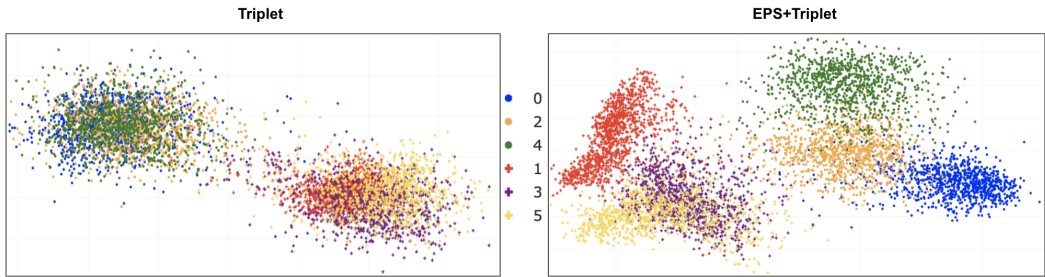

Figure 2: Embedding examples from the MNIST validation set, after training using only even/odd labels. Different colors indicate different digits. **Left:** Using Triplet-loss, class collapsing pushes all intra-class digits to overlapping clusters. **Right:** With EPS, different digits form separate clusters. Retrieval or classification using the odd-vs-even task/metric is more effectively implemented using the embedding on the right, even though the embedding on the left is learned with a loss that more strictly optimizes for the task.

Table 1: Recall@k evaluated on MNIST dataset. The train classes are digits 0-5 and the test classes are digits 6-9

| model | MNIST Train Digits | | | MNIST Test Digits | | |
|---|---|---|---|---|---|---|
| | R@1 | R@5 | R@10 | R@1 | R@5 | R@10 |
| Triplet | 42.01 | 87.51 | 96.56 | 35.16 | 80.86 | 93.26 |
| EPS + Triplet [ours] | **65.78** | **93.57** | **97.38** | **42.31** | **83.86** | **93.61** |

clusters is by using K-means algorithm on the embedding space, with K equal to the number of classes. However, this prevents from the measurement capturing more diverse solutions in which homogeneous clusters appear only when using larger amount of clusters. Regular NMI prefers solutions with class-collapsing. Therefore, we increase the number of clusters in the NMI evaluation (denote it by NMI+) we also report the regular NMI score.

## 5.1 MNIST EVEN/ODD EXAMPLE

To demonstrate the class-collapsing phenomena, we take the MNIST dataset (Lecun et al., 1998), and split the digits according to odd and even. From a visual perspective this is an arbitrary separation. We took the first 6 digits for training and left the remaining 4 digits for testing. We used a simple shallow architecture which result in an embedding function from the image space to $\mathbb{R}^2$ (For implementation details see Appendix C).

We train the network using the triplet loss. We compare our sampling method to random sampling of positive examples (the regular loss). As can be seen in Figure 2, the regular training without EPS suffers from class-collapsing. Training with EPS creates a richer embedding in which there is a clear separation not only between the two-classes, but also between different digits from the same class. As expected, the class-collapsing embedding preforms worse on the test data with the unseen digits, see Table 1.

## 5.2 FINE-GRAINED RECOGNITION EVALUATION

We compare our approach to previous popular sampling methods and losses. The evaluation is conducted on standard benchmarks for zero-shot learning and image retrieval following the common splitting and evaluation practice (Wu et al., 2017; Movshovitz-Attias et al., 2017; Brattoli et al., 2019). We build our implementation on top of the framework of Roth et al., which allow us to have a fair comparison between all the tested methods with an embedding of fix size (128). For more implementation details and consistency of the results, see Appendix C.

### 5.2.1 DATASETS

We evaluate our model on the following datasets.

Table 2: Recall@k and NMI performance on Cars196 and CUB200- 2011. NMI+ indicate the NMI measurement when using 10 (number of classes) clusters. Our EPS method improves in all cases. [†]: Our re-implemented version with the same embedding dimension.

| model | Cars-196 | | | | | CUB-200 | | | | |
|---|---|---|---|---|---|---|---|---|---|---|
| | R@1 | R@2 | R@4 | NMI | NMI+ | R@1 | R@2 | R@4 | NMI | NMI+ |
| Trip. + SH | 51.5 | 63.8 | 73.5 | 53.4 | - | 42.6 | 55.0 | 66.4 | 55.4 | - |
| Trip. + SH[†] | 76.1 | 84.4 | 90.0 | 65.1 | 68.5 | 61.5 | 73.4 | 82.5 | 66.2 | 68.1 |
| ProxyNCA | 73.2 | 82.4 | 86.4 | 64.9 | - | 49.2 | 61.9 | 67.9 | 64.9 | - |
| ProxyNCA[†] | 77.1 | 85.2 | 91.2 | 65.6 | 68.9 | 63.1 | 74.8 | 83.8 | 67.2 | 68.7 |
| Dist-Margin | 79.6 | 86.5 | 91.9 | **69.1** | 70.4 | 63.6 | 74.4 | 83.1 | **69.0** | 68.7 |
| MS | 77.3 | 85.3 | 90.5 | - | - | 57.4 | 69.8 | 80.0 | - | - |
| MS[†] | 81.2 | 89.1 | 93.5 | 60.5 | 71.1 | 62.3 | 73.3 | 82.1 | 59.8 | 68.0 |
| EPS + Trip. + SH | 78.3 | 85.9 | 91.4 | 59.8 | 69.8 | 61.8 | 73.6 | 82.4 | 62.4 | 68.0 |
| EPS + Dist-Margin | **83.6** | **89.5** | **93.6** | 67.3 | **72.4** | **64.7** | **75.2** | **84.3** | 68.2 | **69.4** |
| EPS + MS | 82.9 | 89.4 | 93.2 | 60.0 | 72.0 | 63.3 | 74.2 | 82.5 | 61.2 | 68.2 |

Table 3: Recall@k and NMI performance on Omniglot dataset. In both cases the training was done with only language labels. **Right:** evaluation on language labels. **Left:** evaluation on letter labels. NMI+ indicate the NMI measurement when using 30*(number of classes) clusters. Our EPS method improves in both cases.

| model | Omniglot-letters | | | | | Omniglot-languages | | | | |
|---|---|---|---|---|---|---|---|---|---|---|
| | R@1 | R@2 | R@4 | R@8 | NMI | R@1 | R@2 | R@4 | R@8 | NMI+ |
| Trip. + SH | 49.4 | 60.0 | 69.2 | 76.9 | 66.2 | 71.0 | 80.2 | 87.6 | 92.4 | 38.7 |
| ProxyNCA | 49.1 | 60.4 | 70.9 | 78.9 | 69.0 | 73.0 | 82.1 | 88.8 | 93.5 | 43.3 |
| Dist-Margin | 49.4 | 61.1 | 70.1 | 79.2 | 68.9 | 73.2 | 82.3 | 89.1 | 94.0 | 43.5 |
| MS | 57.7 | 68.5 | 77.3 | 83.8 | 69.2 | 78.8 | 86.4 | 92.0 | 95.4 | 46.0 |
| EPS + Trip. + SH | **68.4** | **79.3** | **86.9** | **92.1** | **79.6** | **85.2** | **91.1** | **94.9** | **97.3** | **52.6** |
| EPS + Dist-Margin | 66.2 | 76.7 | 84.8 | 90.3 | 77.9 | 83.0 | 89.4 | 93.6 | 96.4 | 50.7 |
| EPS + MS | 68.7 | 79.1 | 86.9 | 92.2 | 77.3 | 86.2 | 91.7 | 94.9 | 97.2 | 53.8 |

- **Cars-196** (Krause et al., 2013), which contains 16,185 images of 196 car models. We follow the split in Wu et al. (2017), using 98 classes for training and 98 classes for testing.

- **CUB200-2011** (Wah et al., 2011), which contains 11,788 images of 200 bird species. We also follow Wu et al. (2017), using 100 classes for training and 100 for testing.

- **Omniglot** (Lake et al., 2015), which contains 1623 handwritten characters from 50 alphabet. In our experiments we only use the alphabets labels during the training process, i.e, all the characters from the same alphabet has the same class. We follow the split in Lake et al. (2015) using 30 alphabets for training and 20 for testing.

### 5.2.2 RESULTS

We tested our sampling method with 3 different losses: Triplet (Chechik et al., 2010), Margin (Wu et al., 2017) and Multi-Similarity (MS) (Wang et al., 2019). For the Margin loss experiment, we combine our sampling method with distance sampling (Wu et al., 2017); this could be done because the distance sampling only constrains on the negative samples, where our method only constrains on the positive samples. We set the margin $\alpha = 0.2$ and initialized $\beta = 1.2$ as in (Wu et al., 2017). For the Triplet we combine our method with semi-hard sampling (Schroff et al., 2015) by fixing the positive according to EPS and then using semi-hard sampling for choosing the negative examples. For the MS loss we replace the positive hard-mining method with EPS and use the same hard-negative method. We use the same hyper-paremeters as in (Wang et al., 2019) $\alpha = 2, \lambda = 1, \beta = 50$.

Results are summarized in Tables 2 and 3. We can see that our method achieves the best performance on all tested datasets. It is important to note that in the baseline models, when using Semi-hard sampling, the sampling strategy was done also on the positive part as suggest in the original papers. We see that replacing the semi-hard positive sampling with easy-positive sampling, improve results in all the experiments. The improvement gain becomes larger as the dataset classes can be partitioned

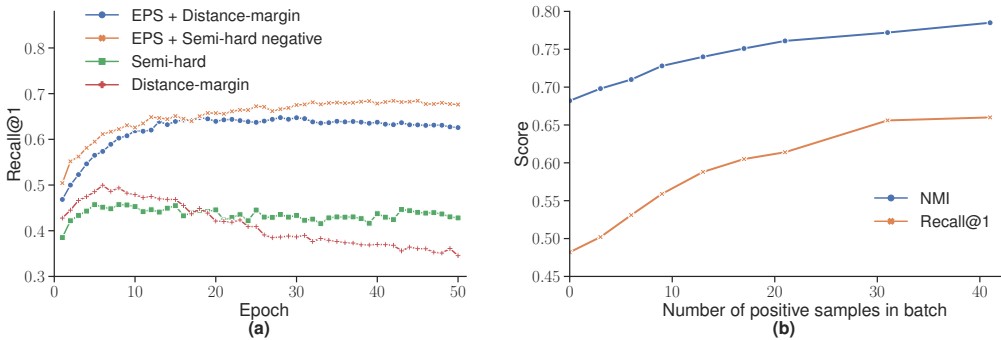

Figure 3: Results on Omniglot-letters. **(a)** Recall@1 performance of each model per epoch. **(b)** performance of *EPS + distance-margin* model on the Omniglot dataset, as a function of the number of positive samples in batch (where zero is equivalent to only using only distance sampling). Increasing the number the number of positive samples enhances the model performance.

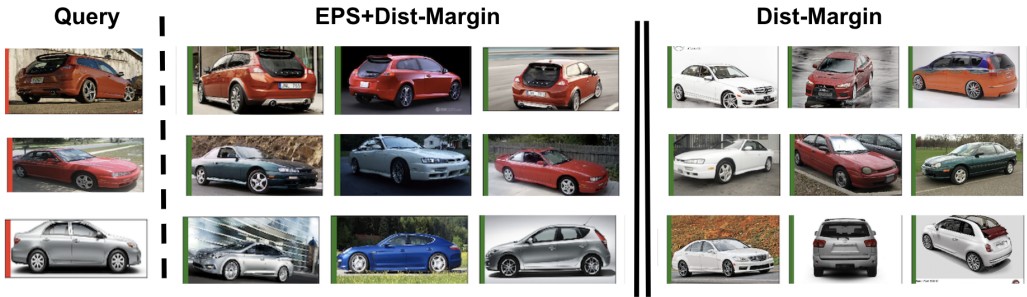

Figure 4: Retrieval results for randomly chosen query images in Cars196 dataset. Using EPS creates more homogeneous neighbourhood relationships with respect to the car viewpoint.

more naturally to a small number of sub-clusters which are visually homogeneous. In Cars196 dataset it is the car viewpoint, where in Omniglot it is the letters in each language. As can be seen in Table 3, using EPS on the Omniglot dataset result in creating an embedding in which in most cases the nearest neighbor in the embedding consists of element of the same letter, although the network was trained without these labels. In Figure 4 we can see a qualitatively comparison of CARS16 models results. EPS seems to create more homogeneous neighbourhood relationships with respect to the the viewpoint of the car. More results and comparisons can be find in Appendix C.

### 5.2.3 Positive batch size effect

An important hyperparameter in our sampling method is the number of positive batch samples, from which we select the closest one in the embedding space to the anchor. If the class is visually diverse and the number of positive samples in batch is low, than with high probability the set of all the positive samples will not contain any visually similar image to the anchor. In case of the Omniglot experiment, the effect of this hyperparameter is clear; It determines the probability that the set of positive samples will include a sample from the same letter as the anchor letter. As can be seen in Figure 3(b), the performance of the model increases as the probability of having another sample with the same letter as the anchor increases.

## 6 Conclusion

In this work we demonstrate the importance of positive sampling strategies when using embedding losses for metric learning. We investigate the class collapsing phenomena with respect to popular embedding losses such as the Triplet loss and the Margin loss. While in clean environments there is a diverse and rich family of optimal solutions, when noise is present, the optimal solution collapses

to a degenerate embedding. We propose a simple solution to this issue based on 'easy' positive sampling, and prove that indeed adding this sampling results in non-degenerate embeddings. We also compare and evaluate our method on standard image retrieval datasets, and demonstrate a consistent performance boost on all of them. While our method and results have been limited to metric learning frameworks, we believe that our sampling scheme will also be useful in other related settings, including supervised contrastive learning, which we leave to future work.

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

APPENDIX

A: PROOFS FOR THE THEOREMS IN SUBSECION 4.2

**Theorem 1.** *Let $f : O \rightarrow \mathbb{R}^m$ be an embedding, which minimize $\mathbb{EO}_{trip}(f)$, then $f$ has the class-collapsing property with respect to all classes.*

*Proof.* Define a new random variables such that for every $1 \leq r_1, r_2 \leq t$:

$$h_{r_1,r_2}(Y,Z) = \begin{cases} 1 & Y = r_1 \wedge Z = r_2 \\ 0 & else \end{cases}$$

observe that

$$\bar{\delta}_{Y_1,Y_2} \cdot (1 - \bar{\delta}_{Y_1,Y_3}) = \sum_{\substack{1 \leq r_1, r_2 \leq t \\ r1 \neq r_2}} \mathbf{1}_{Y_1=r_1} \cdot h_{r_1,r_2}(Y_2, Y_3) = \sum_{\substack{1 \leq r_1, r_2 \leq t \\ r1 \neq r_2}} \mathbf{1}_{Y_1=r_2} \cdot h_{r_1,r_2}(Y_3, Y_2)$$

Since the variables are independent

$$\mathbb{E}(\bar{\delta}_{Y_1,Y_2} \cdot (1 - \bar{\delta}_{Y_1,Y_3})) = \frac{1}{2} \cdot \sum_{\substack{1 \leq r_1, r_2 \leq t \\ r1 \neq r_2}} \mathbb{E}(\mathbf{1}_{Y_1=r_1}) \cdot \mathbb{E}(h_{r_1,r_2}(Y_2, Y_3)) + \mathbb{E}(\mathbf{1}_{Y_1=r_2}) \cdot \mathbb{E}(h_{r_1,r_2}(Y_3, Y_2)).$$

Define: $\bar{D}(x_1, x_2, x_3) := (D_{x_1,x_2} - D_{x_1,x_3} + \alpha)_+$

Rearranging the terms we get

$$n^3 \cdot \mathbb{EO}_{trip}(f) = \sum_{x_1,x_2,x_3 \in X} (\mathbb{E}(\bar{\delta}_{Y_1,Y_2} \cdot (1 - \bar{\delta}_{Y_1,Y_3})) \cdot \bar{D}(x_1, x_2, x_3) =$$

$$\sum_{\substack{x_1,x_2,x_3 \in X \\ 1 \leq r_1 \neq r_2 \leq t}} (\mathbb{E}(\mathbf{1}_{Y_1=r_1}) \cdot \mathbb{E}(h_{r_1,r_2}((Y_2, Y_3)) + \mathbb{E}(\mathbf{1}_{Y_1=r_2}) \cdot \mathbb{E}(h_{r_1,r_2}(Y_3, Y_2))) \cdot \bar{D}(x_1, x_2, x_3) =$$

$$\sum_{\substack{x_1,x_2,x_3 \in X \\ 1 \leq r_1 \neq r_2 \leq t}} (h_{r_1,,r_2}(Y_2, Y_3)) \cdot (\mathbb{E}(\mathbf{1}_{Y_1=r_1}) \cdot \bar{D}(x_1, x_2, x_3) + \mathbb{E}(\mathbf{1}_{Y_1=r_2}) \cdot \bar{D}(x_1, x_3, x_2)) =$$

Therefore, if

$$K(i,j,k,r_1,r_2) = \cdot (\mathbb{E}(\mathbf{1}_{Y_1=r_1}) \cdot \bar{D}(x_i, x_j, x_k) + \mathbb{E}(\mathbf{1}_{Y_1=r_2}) \cdot \bar{D}(x_1, x_k, x_j)),$$

then $\mathbb{EO}_{trip}(f)$ can be written as

$$\mathbb{EO}_{trip}(f) = \frac{1}{n^3} \sum_{\substack{1 \leq i,j,k \leq n \\ 1 \leq r_1 \neq r_2 \leq t}} \mathbb{E}(h_{r_1,r_2}(Y_j, Y_k)) \cdot K(i,j,k,r_1,r_2)$$

For every $x_i \in X$, define:

$$(\mathbb{EO}_{trip}(f))_{x_i} = \frac{1}{n^2} \cdot \sum_{\substack{1 \leq j,k \leq n \\ 1 \leq r_1 \neq r_2 \leq t}} (\mathbb{E}(h_{r_1,r_2}(Y_j, Y_k)) \cdot K(x_i, x_j, x_k, r_1, r_2)$$

Let $f : X \rightarrow \mathbb{R}^m$ be an embedding, fix $1 \leq r \leq t$ and $x_i \in A_r$, $x_j, x_k \in X$ with

$$\| f(x_i) - f(x_j) \| = w, \quad \| f(x_i) - f(x_k) \| = h.$$

By definition:

$$K(i,j,k,r_1,r_2) = \begin{cases} p \cdot (h - w + \alpha)_+ + q(w - h + \alpha)_+ & r_1 = r \wedge r_2 \neq r \\ q \cdot (h - w + \alpha)_+ + p(w - h + \alpha)_+ & r_2 = r \wedge r_1 \neq r \\ p \cdot (h - w + \alpha)_+ + p(w - h + \alpha)_+ & r_1 = r \wedge r_2 = r \\ q \cdot (h - w + \alpha)_+ + q(w - h + \alpha)_+ & r_1 \neq r \wedge r_2 \neq r \end{cases}$$

Since $0 < p < 1$, in order to get minimal $K(i, j, k, r_1, r_2)$ value, $h$ and $w$ must satisfy $|h - w| \leq \alpha$. In this case we have

$$K(i, j, k, r_1, r_2) = \begin{cases} (p+q) \cdot \alpha + (h-w)(p-q) & r_1 = r \wedge r_2 \neq r \\ (p+q) \cdot \alpha + (w-h)(p-q) & r_2 = r \wedge r_1 \neq r \\ 2 \cdot \alpha & r_1 = r \wedge r_2 = r \\ 2 \cdot \alpha & r_1 \neq r \wedge r_2 \neq r \end{cases}$$

Therefore,

$$\sum_{r_2 \in \{1,.r-1,r+1,.t\}} (\mathbb{E}(h_{r,r_2}(Y_j, Y_k)) \cdot K(x_i, x_j, x_k, r_1, r_2) + (\mathbb{E}(h_{r_2,r}(Y_j, Y_k)) \cdot K(x_i, x_j, x_k, r_1, r_2) =$$

$$= (p+q) \cdot \alpha \Big( \sum_{r_2 \in \{1,.r-1,r+1,.t\}} (\mathbb{E}(h_{r,r_2}(Y_j, Y_k)) + (\mathbb{E}(h_{r_2,r}(Y_j, Y_k)))) +$$

$$(h-w)(p-q))\Big( \sum_{r_2 \in \{1,.r-1,r+1,.t\}} \mathbb{E}(h_{r,r_2}(Y_j, Y_k)) \Big) - \mathbb{E}(h_{r_2,r}(Y_j, Y_k))$$

We split to three cases:

1. If $x_j, x_k \in A_r$ or $x_j, x_k \notin A_r$ then: $\mathbb{E}(h_{r,r_2}(Y_j, Y_k)) = \mathbb{E}(h_{r_2,r}(Y_j, Y_k))$. Hence,

$$(h-w)(p-q))\Big( \sum_{r_2 \in \{1,.r-1,r+1,.t\}} \mathbb{E}(h_{r,r_2}(Y_j, Y_k)) \Big) - \mathbb{E}(h_{r_2,r}(Y_j, Y_k)) = 0$$

2. If $x_j \in A_r$ and $x_k \notin A_r$, then $\mathbb{E}(h_{r,r_2}(Y_j, Y_k)) > \mathbb{E}(h_{r_2,r}(Y_j, Y_k))$, therefore

$$(h-w)(p-q))\Big( \sum_{r_2 \in \{1,.r-1,r+1,.t\}} \mathbb{E}(h_{r,r_2}(Y_j, Y_k)) \Big) - \mathbb{E}(h_{r_2,r}(Y_j, Y_k))$$

Since $p > 0.5$ and $|h - w| \leq \alpha$, the minimal value is achieved whenever $h = 0$ and $w = \alpha$.

3. In the same way if $x_k \in A_r$ and $x_j \notin A_r$, then $\mathbb{E}(h_{h_{r_2},r}(Y_j, Y_k)) = \mathbb{E}(h_{r,r_2}(Y_j, Y_k))$ and the minimal value is achieved whenever $h = \alpha$ and $w = 0$.

In conclusion, if $x_i \in A_r$, an embedding $f^*$ satisfies

$$(\mathbb{EO}_{trip}(f^*))_{x_i} = \min\{(\mathbb{EO}_{trip}(f))_{x_i} | f : X \to \mathbb{R}^m\}$$

iff $f^*(x_j) = f^*(x_i)$ for every $x_j \in A_r$, and $\| f^*(x_j) - f^*(x_i) \| = \alpha$ for every $x_j \notin A_r$. □

We will now prove the same theorem with respect to the margin loss.

**Theorem 2.** *Let $f : O \to \mathbb{R}^m$ be an embedding, which minimize*

$$\mathbb{EO}_{margin}(f, \beta) = \frac{1}{n^2} \sum_{x_i, x_j \in X} \mathbb{EL}^f_{margin}(x_i, x_j),$$

*then $f$ has the class-collapsing property with respect to all classes.*

*Proof.* Observe that if $x_i, x_j \in A_r$, then

$$\mathbb{EL}^f_{margin}(x_i, x_j) = p \cdot (D_{x_i, x_j} - \beta_{x_i} + \alpha)_+ + (1 - p) \cdot (\beta_{x_i} - D_{x_i, x_j} + \alpha)_+$$

Since $0 < p < 1$, then the maximal value is achieved whenever $|D_{x_i, x_j} - \beta_{x_i}| \leq \alpha$, in this case:

$$\mathbb{EL}^f_{margin}(x_i, x_j) = (2p - 1) \cdot (D_{x_i, x_j} - \beta_{x_i}).$$

In the same way in case $x_i \in A_r$ and $x_j \notin A_r$ then:

$$\mathbb{EL}^f_{margin}(x_i, x_j) = (2p - 1) \cdot (\beta_{x_i} - D_{x_i, x_j}).$$

Combining both directions we get:

$$\sum_{x_j \in X} \mathbb{EL}^f_{margin}(x_i, x_j) = (2p - 1) \cdot \left( \sum_{Y_j \in A} D_{x_i, x_j} - \sum_{Y_j \notin A} D_{x_i, x_j} \right)$$

Since: $p > 0.5$ and $|D_{x_i, x_j} - \beta_{x_i}| \leq \alpha$, the minimal value is achieved whenever $D_{x_i, x_j} = 0$, $D_{x_i, x_k} = 2\alpha$ and $\beta_{x_i} = \alpha$, for every $x_i, x_j \in A_r$, $x_k \notin A_r$. □

## B: Easy Positive Sampling in noisy environment

In this subsection we analyse the EPS method from the theoretical prospective, using the framework defined in Section 4. We use the same notions as in sections 3 and 4.

Define: $\Phi(y_i, y_j) = \begin{cases} 1 & y_i = y_j \wedge D_{x_i,x_j} = min\{D_{x_i,x_k} \mid y_k = y_i\} \\ 0 & else \end{cases}$. Then, the easy positive sampling loss can be defined by:

$$\frac{1}{n} \sum_{1 \leq i,j,k \leq n} \Phi(y_i, y_j) \cdot \mathcal{L}^f_{trip}(x_i, x_j, x_k)$$

for the triplet loss and

$$\frac{1}{n} \sum_{1 \leq i,j \leq n} \left( \Phi(y_i, y_j) \cdot \mathcal{L}^{f,\beta}_{margin}(x_i, x_j) \right) + 1_{y_i \neq y_j} \mathcal{L}^{f,\beta}_{margin}(x_i, x_j)$$

for the margin loss.

In the noisy environment stochastic case, using section 4 notions, $\Phi$ become a random variable:

$$\bar{\Phi}(Y_i, Y_j) = \begin{cases} 1 & Y_i = Y_j \wedge \forall t \left( (D_{x_i,,x_t} < D_{x_i,x_j}) \rightarrow Y_t \neq Y_i \right) \\ 0 & else \end{cases}$$

Therefore, the triplet loss with EPS in the noisy environment case, become:

$$\mathbb{E}\mathcal{L}^f_{EPStrip}(x_i, x_j, x_k) = \mathbb{E}\left( \bar{\Phi}(Y_i, Y_j) \cdot \bar{\delta}_{Y_i, Y_j} \cdot (1 - \bar{\delta}_{Y_i, Y_k}) \right) \cdot \left( D^f_{x_i,x_j} - D^f_{x_i,x_k} + \alpha \right)_+$$

and for the margin loss with EPS we have:

$$\mathbb{E}\mathcal{L}^f_{EPSmargin}(x_i, x_j) = \mathbb{E}(\bar{\Phi}(Y_i, Y_j) \cdot \bar{\delta}_{Y_i, Y_j}) \cdot (D^f_{x_i,x_j} - \beta_{x_i} + \alpha)_+ + \mathbb{E}(1 - \bar{\delta}_{Y_i, Y_j}) \cdot (\beta_{x_i} - D^f_{x_i,x_j} + \alpha)_+$$

As in section 4.2 we assume that $Y = \{Y_1, .., Y_n\}$ is a set of independent binary random variables. Let $A_1, .., A_t \subset X, 0.5 < p < 1$ such that: $|A_j| = \frac{n}{t}$ and

$$\mathbb{P}(Y_i = k) = \begin{cases} p & x_i \in A_k \\ q' := \frac{1-p}{t-1} & x_i \notin A_k \end{cases}$$

For simplicity we assume that every $1 \leq i \leq \frac{n}{t}$ satisfies $x_{\frac{n \cdot i}{t}+1}, .., x_{\frac{n \cdot i}{t}+t} \in A_i$

We prove first that the minimal embedding with respect to both losses does not satisfy the class collapsing property. Let $f_1$ be an embedding function such that:

$$D^{f_1}_{x_i,x_j} = \begin{cases} 0 & (\exists r)(x_i, x_j \in A_r) \\ \alpha & else \end{cases}$$

and $f_2$ an embedding such that:

$$D^{f_1}_{x_1,x_2} = \begin{cases} 0 & (\exists r)(x_i, x_j \in A_r) \wedge \sim ((i < \frac{t}{2n} \wedge j > \frac{t}{2n}) \vee (i > \frac{t}{2n} \wedge j < \frac{t}{2n}) \\ \alpha & else \end{cases}$$

$f_1$ represent the case of class collapsing, where $f_2$ represent the case where there are two modalities for the first class. In order to show that the minimal embedding does not satisfy the class collapsing property it suffice to prove that

$$\frac{1}{n} \sum_{1 \leq i,j,k \leq n} \mathcal{L}^{f_2}_{EPStrip}(x_i, x_j, x_k) < \frac{1}{n} \sum_{1 \leq i,j,k \leq n} \mathcal{L}^{f_1}_{EPStrip}(x_i, x_j, x_k)$$

and

$$\frac{1}{n} \sum_{1 \leq i,j \leq n} \mathcal{L}^{f_2}_{EPSmargin}(x_i, x_j) < \frac{1}{n} \sum_{1 \leq i,j \leq n} \mathcal{L}^{f_1}_{EPSmargin}(x_i, x_j).$$

**Remark:** For both losses the definition requires a strict order between the elements, therefore by distance zero, we meant infinitesimal close, the order between the elements inside the sub-clusters is random, and element between set $A_1$ are closer then set $A_1^c$ in both embeddings. For simplification we neglect this infinitesimal constants in the proofs.

**Claim 1.** *There exists $M$ such that if $n \geq M$, then:*

$$\frac{1}{n} \sum_{1 \leq i,j,k \leq n} \mathcal{L}^{f_2}_{EPStrip}(x_i, x_j, x_k) < \frac{1}{n} \sum_{1 \leq i,j,k \leq n} \mathcal{L}^{f_1}_{EPStrip}(x_i, x_j, x_k)$$

*Proof.* Fix $x_1$, WOLOG we may assume in both embeddings that $D^{f_j}_{x_1,x_i} < D^{f_j}_{x_1,x_k}$ for every $j \in \{1,2\}$ and $1 \leq i < k \leq n$. It suffice to prove that

$$\frac{1}{n} \sum_{1 \leq j,k \leq n} (\mathcal{L}^{f_1}_{EPStrip}(x_1, x_j, x_k) - \mathcal{L}^{f_2}_{EPStrip}(x_1, x_j, x_k)) > 0$$

Let $q = (1 - p)$, observe that

$$\mathbb{P}(\bigwedge_{1 \leq t < j} Y_i \neq Y_t) = p^{m+1} \cdot q^{j-2-m} + p^{j-2-m} q^{j+1} \leq 2p^{j-1}$$

where $m = |\{t \,|\, t \leq j, \ Y_t \in A_1\}|$. Thus if $j \geq \frac{n}{2t}$, we have

$$\mathcal{L}^{f_2}_{EPStrip}(x_1, x_j, x_k) \leq \mathbb{P}(\bigwedge_{1 \leq t < j} Y_i \neq Y_t) \cdot 2 \cdot \alpha \leq 4 \cdot \alpha p^{j-1}.$$

Therefore,

$$\frac{1}{n} \sum_{j > \frac{n}{2t}, 1 \leq k \leq n} \mathcal{L}^{f_2}_{EPStrip}(x_1, x_j, x_k) \leq \sum_{j > \frac{n}{2t}} 4 \cdot \alpha p^j = 4 \cdot n \cdot \alpha p^{\frac{n}{2t}} \cdot \sum_{j=0}^{\frac{n(2t-1)}{2t}} p^j = 4 \cdot \alpha p^{\frac{n}{2t}} \cdot \frac{1 - q^{n(2t-1)/2t}}{1 - q} \overset{n \to \infty}{\to} 0$$

For $j \leq \frac{n}{2t}$ and $k \leq \frac{n}{2t}$ of $k > \frac{n}{t}$, we have $\mathcal{L}^{f_1}_{EPStrip}(x_1, x_j, x_k) = \mathcal{L}^{f_2}_{EPStrip}(x_1, x_j, x_k)$. Hence, the only case left is $j \leq \frac{n}{2t}$ and $\frac{n}{2t} < k \leq \frac{n}{t}$. In this case: $\mathcal{L}^{f_2}_{EPStrip}(x_1, x_j, x_k) = 0$, where

$$\mathcal{L}^{f_1}_{EPStrip}(x_1, x_j, x_k) = (p^2 \cdot q^{j-1} + q^2 \cdot p^{j-1}) \cdot \alpha \geq q^{j+1} \alpha$$

and we get:

$$\frac{1}{n} \cdot \sum_{j \leq \frac{n}{2t}, \frac{n}{2t} \leq k \leq \frac{n}{t}} \mathcal{L}^{f_1}_{EPStrip}(x_1, x_j, x_k) - \mathcal{L}^{f_2}_{EPStrip}(x_1, x_j, x_k) \geq$$

$$\alpha \cdot q^2 \cdot \sum_{j=0}^{\frac{n}{2t}} q^i = \alpha \cdot q^2 \cdot \frac{1 - q^{n/2t}}{1 - q} \overset{n \to \infty}{\to} \alpha q^2 \cdot \frac{1}{1 - q}$$

Choosing $M$ such that

$$\alpha \cdot q^2 \cdot \frac{1 - q^{M/2t}}{1 - q} > 4 \cdot \alpha p^{\frac{M}{4}} \cdot \frac{1 - q^{M(2t-1)/2t}}{1 - q}$$

will satisfy that for every $n > M$:

$$\frac{1}{n} \sum_{1 \leq i,j,k \leq n} \mathcal{L}^{f_2}_{EPStrip}(x_i, x_j, x_k) < \frac{1}{n} \sum_{1 \leq i,j,k \leq n} \mathcal{L}^{f_1}_{EPStrip}(x_i, x_j, x_k)$$

$\square$

**Claim 2.** *There exists $M$ such that if $n \geq M$ then:*

$$\frac{1}{n} \sum_{1 \leq i,j \leq n} \mathcal{L}^{f_2}_{EPSmargin}(x_i, x_j) < \frac{1}{n} \sum_{1 \leq i,j \leq n} \mathcal{L}^{f_1}_{EPSmargin}(x_i, x_j)$$

*Proof.* For every $1 \leq j \leq \frac{n}{2t}$ or $\frac{n}{t} < j \leq n$ we have:

$$\mathcal{L}^{f_1}_{EPSmargin}(x_i, x_j) = \mathcal{L}^{f_1}_{EPSmargin}(x_i, x_j)$$

For $\frac{n}{2t} < j \leq \frac{n}{2}$:

$$\mathcal{L}^{f_2}_{EPSmargin}(x_i, x_j) = 2 \cdot p \cdot q \cdot \beta_{x_i} + (p^2 q^{j-2} + q^2 p^{j-2}) \cdot (2 \cdot \alpha - \beta_{x_i})$$

while:

$$\mathcal{L}^{f_2}_{EPSmargin}(x_i, x_j) = 2 \cdot p \cdot q \cdot (\beta_{x_i} + \alpha) + (p^2 q^{j-2} + q^2 p^{j-2}) \cdot (\alpha - \beta_{x_i})$$

Since $j > \frac{n}{2t}$ the second therm tend to zero. Therefore, taking M such that

$$2qp > (p^2 q^{\frac{M}{2t}-2} + q^2 p^{\frac{M}{2t}-2})$$

will satisfy that for each $n \geq M$

$$\frac{1}{n} \sum_{1 \leq i,j \leq n} \mathcal{L}^{f_2}_{EPSmargin}(x_i, x_j) < \frac{1}{n} \sum_{1 \leq i,j \leq n} \mathcal{L}^{f_1}_{EPSmargin}(x_i, x_j)$$

$\square$

In the previous two claims we prove that the class collapsing solution is not minimal with respect to both the $EPSmargin$ and the $EPStriplet$. In the following claims we prove that not only it is not the minimal solution, looking locally on the direct effect of the EPS losses on a sample which is not one of the closest elements to to the anchor. We prove that the optimal solution in this case is an embedding in which the distance between the sample to the anchor is equal to the margin hyperparameter.

**Claim 3.** *Let $f$ be an embedding. For every $i$, let $i_1, .., i_n$ be such that $D^f_{x_i,x_{i_1}} < D^f_{x_i,x_{i_2}} < ... < D^f_{x_i,x_n}$, Then there exists $M$ such that for every $j > M$ the minimal embedding for $\mathcal{L}^f_{EPSmargin}(x_i, x_j)$ is achived whenever $D^f_{x_i,x_j} = \beta_{x_i} + \alpha$.*

*Proof.* Fix $x_1$, as in the previous claims we will assume:

$$D^f_{x_1,x_1} < D^f_{x_1,x_2} < ... < D^f_{x_1,x_n}$$

As was prove in in Claim 1 $\mathbb{P}(\bigwedge_{1 \leq t < j} Y_i \neq Y_t) \leq 4p^j$, thus

$$\mathbb{E}(\bar{\Phi}(Y_i, Y_j) \cdot \bar{\delta}_{Y_i,Y_j}) \leq \mathbb{P}(\bigwedge_{1 \leq t < j} Y_i \neq Y_t) \leq 4p^{j-1} \overset{j \to \infty}{\to} 0$$

Since the minimal solution for

$$\mathbb{E}(\bar{\Phi}(Y_i, Y_j) \cdot \bar{\delta}_{Y_i,Y_j}) \cdot (D^f_{x_i,x_j} - \beta_{x_i} + \alpha)_+ + \mathbb{E}(1 - \bar{\delta}_{Y_i,Y_j}) \cdot (\beta_{x_i} - D^f_{x_i,x_j} + \alpha)_+$$

satisfies $|\beta_{x_i} - D^f_{x_i,x_j}| \leq \alpha$, we have:

$$\mathcal{L}^{f_1}_{EPSmargin}(x_1, x_j) = \alpha \cdot (\mathbb{E}(\bar{\Phi}(Y_1, Y_j) \cdot \bar{\delta}_{Y_1,Y_j}) + \mathbb{E}(1 - \bar{\delta}_{Y_1,Y_j})) +$$
$$(D^f_{x_1,x_j} - \beta_{x_1}) \cdot (\mathbb{E}(\bar{\Phi}(Y_1, Y_j) \cdot \bar{\delta}_{Y_1,Y_j}) - \mathbb{E}(1 - \bar{\delta}_{Y_i,Y_j}))$$

Since $\mathbb{E}(1 - \bar{\delta}_{Y_1,Y_j}) \geq 2pq$, there exists $M$ such every $j > M$ satisfies

$$(\mathbb{E}(\bar{\Phi}(Y_1, Y_j) \cdot \bar{\delta}_{Y_1,Y_j}) - \mathbb{E}(1 - \bar{\delta}_{Y_i,Y_j})) < 0$$

Therefore the minimal value is achieved whenever $D^f_{x_1,x_j} = \alpha + \beta_{x_1}$. $\square$

The proof in the EPStriplet loss case is similar.

**Claim 4.** *Let $f$ be an embedding. For every $i$, let $i_1, , .., i_n$ be such that $D^f_{x_i,x_{i_2}} < ... < D^f_{x_i,x_n}$. Then there exists $M$ such that for every $j > M$ the minimal embedding for:*

$$\mathcal{L}^f_{EPStrip}(x_i, x_t, x_{t+j}) + \mathcal{L}^f_{EPStrip}(x_i, x_{t+j}, x_t)$$

*is achieved whenever $D_{x_i,x_{t+j}} = D_{x_i,x_t} + \alpha$.*

*Proof.* Define $K(Y_i, Y_i, Y_k) := \mathbb{E}\left(\bar{\Phi}(Y_i, Y_j) \cdot \bar{\delta}_{Y_i,Y_j} \cdot (1 - \bar{\delta}_{Y_i,Y_k})\right)$. Fixing $x_1$, assuming $D^f_{x_1,x_1} < D^f_{x_1,x_2} < ... < D^f_{x_1,x_n}$, We have:

$$\mathcal{L}^f_{EPStrip}(x_1, x_t, x_{t+j}) + \mathcal{L}^f_{EPStrip}(x_1, x_{t+j}, x_t) = K(Y_1, Y_t, Y_{t+j}) \cdot \left(D^f_{x_1,x_t} - D^f_{x_1,x_{t+j}} + \alpha\right)_+$$
$$+ K(Y_1, Y_{t+j}, Y_t) \cdot \left(D^f_{x_1,x_{t+j}} - D^f_{x_1,t} + \alpha\right)_+$$

As in the previous claim, the minimal value is achieved whenever $|D^f_{x_1,x_{t+j}} - D^f_{x_1,x_t}| \leq \alpha$ in this case:

$$\mathcal{L}^f_{EPStrip}(x_1, x_t, x_{t+j}) + \mathcal{L}^f_{EPStrip}(x_1, x_{t+j}, x_t) = \alpha \cdot (K(Y_1, Y_t, Y_{t+j}) + K(Y_1, Y_{t+j}, Y_t))Y_t)) +$$
$$(D^f_{x_1,x_t} - D^f_{x_1,x_{t+j}}) \cdot (K(Y_1, Y_t, Y_{t+j}) - K(Y_1, Y_{t+j}, Y_t))$$

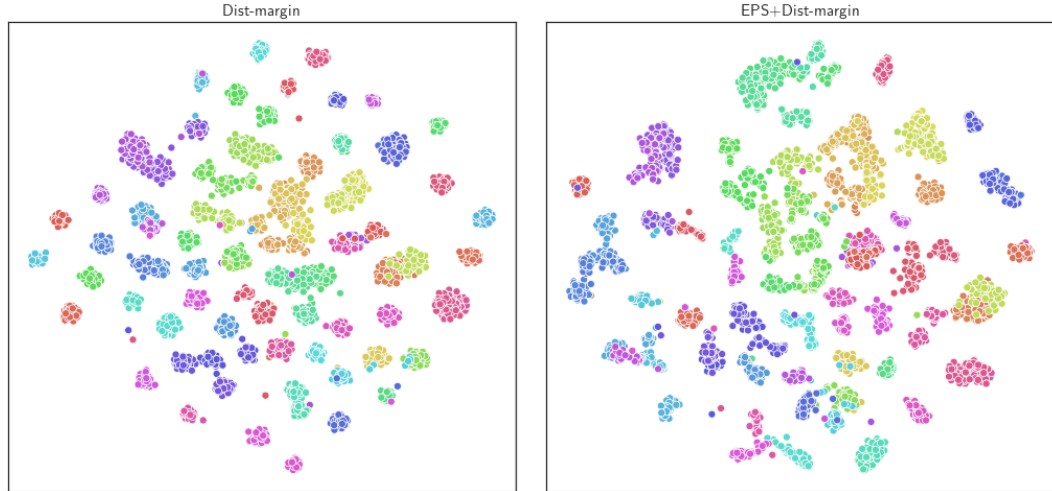

Figure 5: t-SNE visualization of Cars196 training classes (each class has a different color). Training with EPS results in more diverse classes appearance.

Table 4: Results of semi-hard with/without EPS on the Omniglot training dataset. Without EPS the network feet almost perfectly to the training set. However, using EPS results in batter performances on the letters fine-grained task.

|  | Language | | Letters | |
|---|---|---|---|---|
|  | Semi-hard | Semi-hard+EPS | Semi-hard | Semi-hard+EPS |
| NMI | **93.6** | 67.3 | 78.4 | **87.1** |
| R@1 | **99.9** | 94.5 | 70.3 | **77.5** |
| R@2 | **100** | 96.8 | 80.4 | **86.3** |
| R@4 | **100** | 98.1 | 87.9 | **92.4** |
| R@8 | **100** | 99.2 | 93.3 | **96.0** |

On the one hand: $K(Y_1, Y_t, Y_{t+j}) = (\prod_{i \in \{1,2,...,t,t+j\}} p^{t_i} q^{1-t_i}) + (\prod_{i \in \{1,2,...,t,t+j\}} p^{1-t_i} q^{t_i}) \geq q^{t+1}$ where $t_k = \begin{cases} 1 & Y_k \notin A \\ 0 & else \end{cases}$ for $k \in \{2,..,t-1,t+j\}$ and $t_k = \begin{cases} 1 & Y_k \in A \\ 0 & else \end{cases}$ for $k \in \{1,t\}$. On the other hand $K(Y_1 Y_{t+j}, Y_t) \geq Prob(\bigwedge_{1 \leq k < t+j} Y_1 \neq Y_k) \leq 4p^{t+j-1}$. Taking $j$ large enough such that $q^{t+1} \leq 4p^{t+j-1}$, we have:

$$(\mathbb{E}\left(\bar{\Phi}(Y_1, Y_t) \cdot \bar{\delta}_{Y_1, Y_t} \cdot (1 - \bar{\delta}_{Y_1, Y_{t+j}})\right) - \mathbb{E}\left(\bar{\Phi}(Y_1, Y_{t+j}) \cdot \bar{\delta}_{Y_1, Y_{t+j}} \cdot (1 - \bar{\delta}_{Y_1, Y_t})\right)) > 0$$

therefore in such case the minimum is archived whenever $D^f_{x_1, x_{t+j}} = D^f_{x_1, x_t} + \alpha$.

$\square$

## C: More experiments and implementation details

### Embedding behavior on training sets

The class-collapsing phenomena also occur in the training process of the image retrieval datasets. Figure 5 visualise the t-SNE embedding (van der Maaten & Hinton, 2008) of Cars196 training classes. As can be seen, when training without EPS each class fits well to a bivariate normal-distribution with small variance and different means. Training with EPS result in more diverse distributions and in some of the classes fits batter to a mixture of multiple different distributions.

This can also be measured qualitatively on the Omniglod detest; although training without the EPS results in batter overfitting to training samples, the results on the letters fine-grained task are significantly inferior comparing to training with the EPS (Table 4). It is also important to note the low NMI score when using EPS with the number of clusters equal to the number of languages, and the increment of this score when increasing the

Table 5: Std of Recall@1 results. Each model was trained 8 times with different random seeds.

| dataset | model | Without EPS std | With EPS std |
|---|---|---|---|
| cars196 | Margin | 0.17 | 0.27 |
| cars196 | MS | 0.24 | 0.29 |
| cars196 | Trip+SH | 0.20 | 0.47 |
| cub200 | Margin | 29.8 | 0.33 |
| cub200 | MS | 0.43 | 0.36 |
| cub200 | Trip+SH | 0.52 | 0.35 |
| Omniglot-letters | Margin | 0.73 | 0.58 |
| Omniglot-letters | MS | 0.52 | 0.71 |
| Omniglot-letters | Trip+SH | 0.34 | 0.61 |

Table 6: Results of Multi-similarity loss with Embedding size 512 (as in Wang et al. (2019)). Using EPS improve results in both cases.

| | Cars196 | | CUB200 | |
|---|---|---|---|---|
| | MS | MS+EPS | MS | MS+EPS |
| R@1 | 84.1 | **85.5** | 65.7 | **66.7** |
| R@2 | 90.4 | **90.7** | 77.0 | **77.2** |
| R@4 | 94.0 | **94.3** | 86.3 | **86.4** |
| R@8 | 96.5 | **96.7** | **91.2** | 90.9 |

number of clusters to the number of letters. This indicates that training with EPS results in more homogeneous small clusters, which are more blended in the embedding space comparing to training without EPS.

## MNIST ARCHITECTURE DETAILS

For the MNIST even/odd experiment we use a model consisting of two consecutive convolutions layer with (3,3) kernels and 32,64 (respectively) filter sizes. The two layers are followed by Relu activation and batch normalization layer, then there is a (2,2) max-pooling follows by 2 dense layers with 128 and 2 neurons respectively.

## RECOGNITION DATASETS ARCHITECTURE DETAILS

We use an embedding of size 128, and an input size of 224X224 for the first two datasets, and 80X80 for the Omniglot dataset. For all the experiments we used the original bounding boxes without cropping around the object box. As a backbone for the embedding, we use ResNet50 (He et al., 2016) with pretrained weights on imagenet. The backbone is followed by a global average pooling and a linear layer which reduces the dimension to the embedding size. Optimization is performed using Adam with a learning rate of $10^{-5}$, and the other parameters set to default values from Kingma & Ba (2014).

## STABILITY ANALYSIS

Following Musgrave et al. (2020); Fehervari et al. (2019), it was important to us to have a fair comparison between all tested models. Therefore, for all the experiments we use the same framework (Roth et al.), with the same architecture and embedding size (128). We also did not change the default hyper-parameters in all tested methods. We run each experiment 8 times with different random seeds, the reported results are the mean of all the experiments. The std of the Recall@1 results of all experiments can be seen in Table 5. In all cases the differences between the results with and without the EPS are significance.

## MULTI-SIMILARITY COMPARISON

From our experiments, the Multi-similarity loss is highly affected by the batch size. Using Resnet50 backbone, we restrict the number of batch size to 160 for all tested model, which cause to the inferior results of the multi-similarity loss comparing to other methods. For the sake of completeness we provide the results also on inception backbone with embedding size of 512 as in Wang et al. (2019), and batch size of 260. As can be seen in Table 6, also in these cases the results improve when using EPS instead of semi-hard sampling on the positive samples.

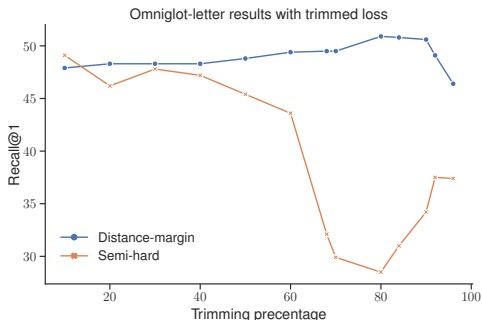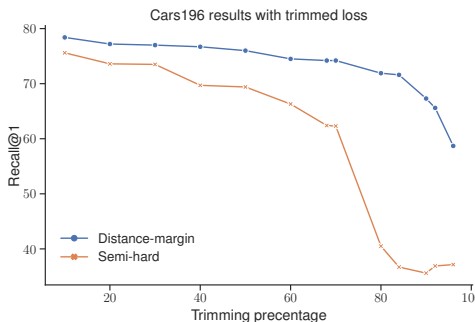

Figure 6: Recall@1 performance with Trimmed loss across varying trimming percentage. Except for small improvement in the Distance-margin case on the Omniglot dataset, in all other cases there is no improvement when applying the Trimmed loss.

TRIMMED LOSS COMPARISON

The situation where a class consists of multiple modes can also be seen as a noisy data scenario with respect to the embedding loss, where positive tuples consisting of examples from different modes are considered as 'bad' labelling. One approach to address noisy labels is by back-propagating the loss only on the k-elements in the batch with the lowest current loss (Shen & Sanghavi, 2019). Although this approach resembles Malisiewicz & Efros (2008), the difference is that in Malisiewicz & Efros (2008) they apply the trimming only on the positive tuples. We test the effect of using Trimmed Loss on random sampled triplets with different level of trimming percentage. As can be seen in Figure 6, there is only a minor improvement when applying the loss on top of the distance-margin loss on the Omniglot-letters dataset. This emphasizes the importance of constraining the trimming to the positive sampling only.

