# OpenReview forum: "Reducing Class Collapse in Metric Learning with Easy Positive Sampling"
_ICLR.cc/2021/Conference — Reject_

### Official Review · AnonReviewer3 · 2020-10-25
**REDUCING CLASS C OLLAPSE IN M ETRIC L EARNING WITH EASY POSITIVE S AMPLING**

**Rating:** 4
**Confidence:** 5

**Review:**

Post-rebuttal: The rebuttal partly addresses my concerns, so I would like to change my score to 4.
------------------------------------------------------
This paper proposes an easy positive sampling method for deep metric learning which aims to reduce the class collapse problem which is found to harm the performance of existing DML methods.

Pros:
1. This paper is well-written and easy to follow.
2. The idea is simple but makes good sense. The author also provide solid theoretical analysis of the flaw of existing methods and the advantage of the proposed easy positive sampling strategy.

Cons:
1. The idea of sampling easy positive for deep metric learning is actually not new. [1] already proposed an easy positive sampling method and the motivation is quite similar (to relax the constraints of intra-class variations). [1] should be cited in this paper.
2. The authors only provide theoretical analysis on the binary case and claims it can be easily extended to the multi-label case, which I find not trivial.
3. A concern is the limited batch size, which might cause the easy positive sample of one particular sample at different iterations to be different (and possibly from different subcluster). This might lead to inconsistent effect of pushing the same sample to different subclusters.
4. Similar to the last one, a more general problem is the theoretical analysis only consider the optimal situation but neglects the nature of batch-based training, which might bring unexpected problems.
5. For the experiments, the performance improvement using the proposed easy positive sampling is not strong. Specifically, the best performance on the Cars196 and CUB200 dataset is achieved with EPS + margin, but the authors did not report the performance of margin loss  with distance-weighted sampling. Comparisons with other sampling methods on the same loss should be provided.
6. The authors should design an experiment to better demonstrate the class collapsing problem on a regular dataset like CUB. The toy experiment on the MNIST dataset is not convincing.

In summary, I think this paper is solid and well-motivated, but I find the idea not new and the experiments not satisfying. The latter weighs more in my decision.

[1] Xuan H, Stylianou A, Pless R. Improved embeddings with easy positive triplet mining[C]//The IEEE Winter Conference on Applications of Computer Vision. 2020: 2474-2482.

---

> ### Author Response · Authors · 2020-11-18
> **Response to Reviewer3**
>
> We thank the reviewer for the review and comments. We provide answers to the comments below.
> 1) In our work, in general, we had a theoretical analysis on two types of losses: (1) The losses that trivially (always) lead to class collapse properties, and (2) The losses that are theoretically robust to class collapse when certain conditions (non-noisy setting) are met. For the former ones, we only briefly discussed them in the paper as from a theoretic perspective it is clear that EPS should resolve class collapsing; for the latter ones, it turns out that noisy environment modelling batter fit the real-world data, thus as our theoretical and extensive experimental analysis demonstrate, this family of losses also need EPS to resolve the class collapsing, and indeed EPS yields improvement with respect to different datasets, losses and negative sampling methods. The analysis and remedy on loss type 2) is the core contribution of our work.
> 	The authors in [1] strived to improve the (N+1)-tuplet loss [2] (eq. 3)  and a few of its variations. The (N+1)-tuplet loss (as the contrastive loss which was discussed in section 3), belongs to loss type (1). Note that these type (1) kinds of losses result in a class collapse in both non-noisy and noisy settings. Therefore we do not believe that the reference interfered with the contribution of our work.
> Regardless, we thank the reviewer for bringing up the relevant reference, we have added it to the reference list and modified the related work section to reflect the discussion.
> 2) We changed the definitions in subsection 4.2 to the multi-class case and modified the theorems and proofs accordingly. This required only small modifications (mainly in the proof of theorem 1), please see the new version.
> 3) The concern on the batch size is addressed in subsection 5.2.3. As we demonstrate on the Omniglot dataset, if the number of inner-class modalities is extremely high (compared to the batch size), then with low probability there will be two samples from the same modality in the mini-batch, and the EPS becomes less effective. However, as can be seen in Figure 3(b), even when the probability is low (less than 10%), there is still a significant improvement when using EPS.
> 4) The triplet and margin loss functions are convex with respect to the distances in the embedding between every two samples. We proved (Theorem 1+2)  that they had a unique global solution (as a function of the distances) in the noisy-environment setting. Therefore if the optimization is done with respect to the distances, then with sufficiently small updates the process should converge to the class-collapsing solution. It is true that in practice the model hypothesis space is restricted, and we are not directly optimizing the distances relations but rather the network parameters. However, These assumptions and theoretical simplification is common, see for example [3] (sec 4.2).
> Please note that in our work, we also have the experiments section, as discussed in the paper, the empirical evidence supporting the theoretical findings.
> 5) It is important to us to clarify that the comparison in all tested experiments with/without EPS was done in the exact same setting (including hyper-parameters like batch-size and even random seeds), the only difference is the positive sampling method. The same applies to the distance-margin experiment. In both cases distance sampling was used on the negative part. We think that the reviewer might be confused about the experiment abbreviation in the tables. To clarify it we changed the abbreviation name in the new draft.
> 6) We add in Appendix C a small subsection describing the embedding behaviour on the training sets. This includes also a t-SNE visualization of Cars-196 which indicates on class collapsing when training without EPS. Another important quantitative indication to the embedding behaviour is the change in performances of the NMI score when increasing the number of clusters. This indicates that training with EPS results in more homogeneous small clusters, which are more blended in the embedding space (compared to training without EPS).
>
> [1] Xuan H, Stylianou A, Pless R. Improved embeddings with easy positive triplet mining[C]//The IEEE Winter Conference on Applications of Computer Vision. 2020: 2474-2482.
> [2] Kihyuk Sohn, “Improved Deep Metric Learning with Multi-class N-pair Loss Objective”
> [3]  Goodfellow et al., “Generative Adversarial Nets”

---

### Official Review · AnonReviewer1 · 2020-10-28
**A simple and effective sampling method, but is used before and its effectiveness is yet to be validated**

**Rating:** 5
**Confidence:** 5

**Review:**

This paper proposes/adopts a simple positive sampling scheme in metric learning: only sampling the easiest positive for each anchor. Authors give a theoretical analysis of how the proposed sampling scheme can reduce class collapse. Experiments on fine-grain retrieval datasets show the effectiveness of the sampling scheme. Using the sampled easiest positive, nearly all current metric learning methods got improved performance.

Pros:

1. propose/adopt a simple easiest positive sampling scheme, and show its usefulness in fine-grain retrieval task;
2. Extensive theoretical analysis of how the proposed sampling scheme can reduce class collapse.

Cons:

1. I don't think this easiest positive sampling scheme is a contribution, though authors give a theoretical analysis of why this scheme can reduce class collapse. Specifically,  Arandjelovic et al. (2016) used exactly the same easiest positive sampling scheme. Though authors explicitly show the difference between Arandjelovic et al. (2016) and the proposed method (Equation in section 4.3), I found no difference.

In the paper of Arandjelovic et al. (2016), they don't clean the positive set (only minor negatives could be included) for efficient training. This is good for practical usage.

2. For section 4, it is good to analyze the Class-collapsing property. However, I would suggest using three classes to derive theorems, rather than using two classes. As a metric-learning problem usually has multiple classes, having two or three classes are usually different stories.

3. While I trust the effectiveness of the easiest positive sampling scheme in the fine-grain image retrieval datasets, I strongly suspect its effectiveness in a broad image retrieval task.

For example, in the following paper:

Radenović, Filip, Giorgos Tolias, and Ondřej Chum. "CNN image retrieval learns from BoW: Unsupervised fine-tuning with hard examples." European conference on computer vision. Springer, Cham, 2016.

This easiest positive sampling scheme falls short in performance.

Combining the conclusions from the above paper and the paper under review, I would say this easiest positive sampling scheme has limited contribution, as it is not broadly applicable.

---

> ### Author Response · Authors · 2020-11-18
> **Response to Reviewer1**
>
> We thank the reviewer for the review, comments, and constructive feedback. We provide answers to the comments below.
> 1) We believe that [1] fundamentally differs with our work in terms of the motivation and problem setting. In [1] the authors collected a noisy dataset, in which every image had: 1) a negative set containing samples of different categories, and 2) a potentially positive set containing at least one sample of the same category (Sec.4, 3rd paragraph). Therefore, they formulate the task in a multiple instance learning setting (Sec.4, second to the last paragraph) and propose an algorithm to perform weakly-supervised metric learning on a noisy positive set. On the other hand, our motivation derived from the problem of class collapse due to intra-class multi-modality in fully-supervised metric learning. In our problem setting samples in the positive set are universally positive samples in the positive set are.
> It can also be inferred that the aim of NetVLAD is not about learning intra-class sub-clusters; on the contrary, the authors in [1] argued that to achieve good retrieval performance a model needed to be invariant w.r.t. perspectives, seasons and lighting, etc. (e.g., Fig. 4 and Sec.4 in [1]). That is to say, NetVLAD aimed to blend all intra-class invariances while we strived to preserve them. Fig. 11 in [1] also strongly supports our argument, in which nighttime images were provided and the authors believed that their method outperformed the baseline by retrieving daytime images of the same category, whereas our method would have retrieved nighttime images of the same category. We believe that the algorithms behave significantly differently because their dataset lacks clear distinct class-modalities due to the incompleteness of the data and a large amount of noise (Sec.4, 1st paragraph).
> 2)  We thank the reviewer for this suggestion, we changed the definitions in subsection 4.2 to the multi-class case and modified the theorems and proofs accordingly. This required small modifications (mainly in the proof of theorem 1), see the new draft.
> 3) The paper addresses the issue of class collapsing and the EPS method was suggested to resolve this issue. In cases where class collapsing is not a concern, for example in Stanford Online Products and In-shop datasets, where the numbers of elements per class is very small, there is no benefit in using EPS. In these cases (as expected) there was no statistical significance when using and not using EPS. However, the scenario of inner-class multi-modalities is broad and happens naturally in many use cases. This is especially true in real-world datasets, in which fine-grained annotations are costly. It is also important to note that this issue was also addressed by others, for example, see [2] in the classification context, which strengthens our belief that this is indeed an important problem by its own.
>
> Q: in [3] This easiest positive sampling scheme falls short in performance.
>
> In [3] they got better results compared to [1]. However, they used external information, i.e., the camera position and the 3D-model  (Sec.4.1, 2nd paragraph), in order to relax the loss constraints on the positive samples relations. With EPS the relaxation is done in an unsupervised way without any external information. It is expected to get a boost in performances when adding labels and doing the sampling in a supervised way, but this is a different problem setting.
>
>
> [1] Arandjelovic et al., “NetVLAD: CNN architecture for weakly supervised place recognition
> [2] Qian et al., “SoftTriple Loss: Deep Metric Learning Without Triplet Sampling”
> [3] Radenović, Filip, Giorgos Tolias, and Ondřej Chum. "CNN image retrieval learns from BoW: Unsupervised fine-tuning with hard examples." European conference on computer vision. Springer, Cham, 2016.

---

### Official Review · AnonReviewer2 · 2020-10-29
**Gap between theory and empirical**

**Rating:** 6
**Confidence:** 4

**Review:**

Motivated by the theoretical results on class collapse problems, this paper proposes a simple positive sampling mechanism called EPS for metric learning. The method is simple -- each sample selects its nearest same-class counterpart in a batch as the positive element. The authors provide both theoretical motivations and empirical studies on the proposed method.

Strengths:
+ Theoretical analysis on the existing class collapse problem for triplet and margin loss
+ well-motivated and simple solutions that are proven to be effective in theory

Weaknesses:
- there is a gap between theoretical analysis and empirical studies. In the analysis, the paper shows in the noisy label setting, margin and triplet loss also induce the class collapse problem but in the empirical study, the paper only conducted analysis on dataset with clean labels.
- the theoretical analysis based on the assumption that the function f can approximate any functions. However for any fixed deep nets, it does not satisfy the requirement

Overall, the EPS method is simple and supported by theoretical analysis. I would find it more convincing if the paper can provide empirical analysis on noisy labeled data. In addition, I wonder how different architectures can affect the difference between previous methods and EPS.

---

> ### Author Response · Authors · 2020-11-18
> **Response to Reviewer2**
>
> We thank the reviewer for the review, comments, and constructive feedback. We provide answers to the comments below.
>
> The two theoretical concerns that were raised by the reviewer are strongly related.
> The noisy environment setting, not only described the aleatoric uncertainty (which is indeed less relevant in the case of clean datasets), but it can also describe the approximation uncertainty which is a result of the model incapacity to perfectly overfit the data. In this case, the probability of the labels can be taken as the results of the Bayes optimal model within the hypothesis space.
>
> Regarding the approximation concern of the reviewer (comment 2);  the approximation uncertainty in deep neural networks is considered to be negligible [1],[2].  However, as we prove in our work (Theorem 1+2), as long as the probability is not exactly 0/1, even a small amount of noise causes the family of optimal solutions to degenerate to only class-collapsing embeddings.
>
> Q: "In addition, I wonder how different architectures can affect the difference between previous methods and EPS."
>
> In appendix C we provide more results on the MS-loss (which is considered to have the best performances among tested losses [3]), with different architecture; An inception backbone with different embedding size (512). This was the architecture that was used in the original paper [4] and achieved the best result.  We reproduce the result from [4] (without EPS) and demonstrate that also in this case there is a significant improvement when using EPS.
>
>
> [1] Cybenko ,  “Approximation by superpositions of a sigmoidal function”
> [2] Tagasovska et al., “Single-Model Uncertainties for Deep Learning”
> [3] Musgrave et al., “A Metric Learning Reality Check”
> [4]  Wang et al, “Multi-Similarity Loss with General Pair Weighting for Deep Metric Learning”

---

> > ### Comment · AnonReviewer2 · 2020-11-24
> > **Response**
> >
> > Thanks to the authors for providing the response and references. It is true that under some conditions, by modifying the architecture (e.g. increasing the width) of the deep net, it can approximate any functions within epsilon distance. However in reality we have the deep net fixed, which means it might be far away from the optimal solution. Regarding the experimental set up, the authors should still provide some experiments on standard benchmark (e.g CUB) with different noise levels on labels so researchers can better understand the method and its advantages. This will also make the experiments section more convincing.

---

### Official Review · AnonReviewer4 · 2020-10-30
**A simple sampling manner for diverse and distinct sub-classes in metric learning**

**Rating:** 6
**Confidence:** 5

**Review:**

The authors find that the popular triplet loss will force all same-class instances to a single center in a noisy scenario, which is not optimal to deal with the diverse and distinct sub-classes. After some analyses, the authors propose a simple sampling strategy, EPS, where anchors only pull the most similar instances. The method achieves good visualization results on MNIST and gets promising performance on benchmarks.

Avoiding class collapse is meaningful and important in metric learning when dealing with some tasks. The analyses in the paper provide insights. Here are some possible issues of this paper.
1. The authors should discuss when we need to avoid such class collapse. Maybe in some cases, pulling all similar instances to a single point leads to more discriminative embeddings. Even some methods are designed following that consideration. Some examples and demonstrations are required.
2. It's better to write a sketch of the analysis on how to extend it to multi-class cases and analyze will the definition of the noise influence the final results.
3. Maybe the authors need to find another real-world dataset with multiple meanings in one class and show the advantage of the proposed method. We can find the improvement of performance on the benchmarks, but the numbers are hard to illustrate the effect of the method.

---

> ### Author Response · Authors · 2020-11-18
> **Response to Reviewer4 comments**
>
> We thank the reviewer for the review, comments, and constructive feedback. We provide answers to the comments below.
> 1) As discussed in the first paragraph of section 3, the standard evaluation and prediction method for image retrieval tasks are typically based on the properties of K-nearest neighbours in the embedding space. In this case, the class-collapsing property is a side-effect which harms the performance, as was demonstrated in the experiments. However, we do acknowledge that under certain circumstances, class-collapsing of the embedding might be a desired property. In the experiment section (1st paragraph) we provide one such metric: the NMI score where the number of clusters equals the number of classes. As discussed in the paper, with respect to this specific metric, splitting the class cluster to separate smaller homogeneous sub-clusters in the embedding space is not desired property, and indeed as was shown in Table 2, using EPS reduced the NMI score with respect to all tested losses and datasets.
> We note that forcefully mapping samples of various modalities into a single cluster (class collapsing) requires memorization of the intra-class relations, which will ultimately hurt generalization.
> 2) We changed the definitions in subsection 4.2 to the multi-class case and modified the theorems and proofs accordingly. This required small modifications (mainly in the proof of theorem 1), please see the revision.
> 3)  In order to batter illustrate the effect of the method, we add in Appendix C a subsection describing the embedding behaviour on the training sets. This includes a visualization on Cars-196 training embedding and results on the Omniglot fine-tuned task, both of them demonstrate the effect on the class-collapsing when training with/without EPS

---

### Decision · Program_Chairs · 2021-01-07
**Final Decision**

**Decision:**

Reject

**Comment:**

This paper is truly borderline. On one hand, the theoretical contribution seems novel and interesting, however, there appears to be somewhat of a gap between theory and practice.

There is unfortunately another problem. According to the authors, the main contribution of this publication is arguably the introduction of the nearest neighbor as the positive example in the triplet loss. However, the authors seem to be unaware of the history of the triplet loss. It was originally introduced by Schultz & Joachims 2004 as a loss over all triplets.  Weinberger et al. 2005 changed it and use the nearest neighbor as "target neighbor", which is called "easy positives" here, as the objective of LMNN. In 2009 Chechik et al. subsequently relaxed this positive neighbor formulation to any similarly labeled sample (going back to the Schultz & Joachims formulation) but sampling triplets. The re-introduction of the nearest neighbor as "easy positive" was then covered by Xuan et al. 2020.

Unfortunately all of this diminishes the novelty significantly and it is clear that the paper in its current form does not have a strong enough contribution. I do encourage the authors to take a close look at the original LMNN publication and Xuan et al and write an improved re-submission for the next conference that maybe focuses more on the theoretical contribution.
Good luck,

AC